# Circulating small extracellular vesicle RNA profiling for the detection of T1a stage colorectal cancer and precancerous advanced adenoma

Li Min[1,2]*, Fanqin Bu[1], Jingxin Meng[2], Xiang Liu[3], Qingdong Guo[1], Libo Zhao[3], Zhi Li[3], Xiangji Li[4], Shengtao Zhu[1]*, Shutian Zhang[1]*

[1]Department of Gastroenterology, Beijing Friendship Hospital, Capital Medical University, State Key Laboratory of Digestive Health, National Clinical Research Center for Digestive Diseases, Beijing Digestive Disease Center, Beijing Key Laboratory for Precancerous Lesion of Digestive Disease, Beijing, China; [2]Key Laboratory of Bio-inspired Materials and Interfacial Science, Technical Institute of Physics and Chemistry, Chinese Academy of Sciences, Beijing, China; [3]Echo Biotech Co., Ltd, Beijing, China; [4]Department of Retroperitoneal Tumor Surgery, International Hospital, Peking University, Beijing, China

*For correspondence:
minli@ccmu.edu.cn (LM);
shengtaozhu@126.com (SZ);
zhangshutian@ccmu.edu.cn (SZ)

**Abstract** It takes more than 20 years for normal colorectal mucosa to develop into metastatic carcinoma. The long time window provides a golden opportunity for early detection to terminate the malignant progression. Here, we aim to enable liquid biopsy of T1a stage colorectal cancer (CRC) and precancerous advanced adenoma (AA) by profiling circulating small extracellular vesicle (sEV)-derived RNAs. We exhibited a full RNA landscape for the circulating sEVs isolated from 60 participants. A total of 58,333 annotated RNAs were detected from plasma sEVs, among which 1,615 and 888 sEV-RNAs were found differentially expressed in plasma from T1a stage CRC and AA compared to normal controls (NC). Then we further categorized these sEV-RNAs into six modules by a weighted gene coexpression network analysis and constructed a 60-gene t-SNE model consisting of the top 10 RNAs of each module that could well distinguish T1a stage CRC/AA from NC samples. Some sEV-RNAs were also identified as indicators of specific endoscopic and morphological features of different colorectal lesions. The top-ranked biomarkers were further verified by RT-qPCR, proving that these candidate sEV-RNAs successfully identified T1a stage CRC/AA from NC in another cohort of 124 participants. Finally, we adopted different algorithms to improve the performance of RT-qPCR-based models and successfully constructed an optimized classifier with 79.3% specificity and 99.0% sensitivity. In conclusion, circulating sEVs of T1a stage CRC and AA patients have distinct RNA profiles, which successfully enable the detection of both T1a stage CRC and AA via liquid biopsy.

## eLife assessment

This study presents a **useful** description of RNA in extracellular vesicles (EV-RNAs) and highlights the potential to develop biomarkers for the early detection of colorectal cancer (CRC) and precancerous adenoma (AA). The data were analysed using overall **solid** methodology and would benefit from further validation of predicted lncRNAs and biomarker validation at each stage of CRC/AA to evaluate the potential application to early detection of CRC and AA.

## Introduction

With 1,880,725 new cases and 915,880 deaths, colorectal cancer (CRC) ranks as the 3rd most commonly diagnosed and 2nd most lethal cancer worldwide in 2020 (*Sung et al., 2021*). Both the incidence and mortality of CRC in developing countries are gradually increasing due to rapid social-economic improvement (*Chen et al., 2016*). Even though a remarkably high CRC incidence was observed in developed areas, such as Europe and North America, the mortality of CRC has dramatically decreased over the past decade owing to the continuous improvement of multidisciplinary treatment and the promotion of CRC screening programs (*Siegel et al., 2017*). Noticeably, endoscopic treatments enable radical resection of T1a stage CRC without open surgery, and the 5-year survival rate of those patients has been improved to over 90% (*Li et al., 2019*; *Cao et al., 2018*). Advanced adenoma (AA) is the crucial precancerous lesion of CRC, which is also resectable under endoscopy (*Hsu et al., 2020*; *Good et al., 2015*; *Tehranian et al., 2020*; *Shaukat et al., 2019*). About 40% of AA patients could progress to invasive CRC within 10 years, which provided a very wide window of opportunity for curative treatment (*Brenner et al., 2007*). Unfortunately, most AA patients were asymptomatic, and hard to be identified by opportunistic screening. Thus, early diagnosis of CRC and precancerous AA is of vital importance.

Endoscopy is the main approach for the screening of CRC and AA, whereas the invasiveness and complicated operation procedures largely restricted its application in asymptomatic populations. For most endoscopy screening programs, participants are simply pre-selected by their age and other epidemiological factors, resulting in low compliance and unsatisfactory cost-effectiveness (*Force, 2016*; *Sung et al., 2015*). Fecal occult blood test (FOBT) and fecal immunohistochemistry test (FIT) are also popular in identifying individuals of high CRC/AA risk, whereas their sensitivities are far from satisfactory (*Haug et al., 2011*). The performances of FOBT and FIT to detect AA are especially disappointing, which exhibited a sensitivity of 20% (95% CI 6.8% to 40.7%) and 32% (95% CI 14.9% to 53.5%), respectively (*Graser et al., 2009*). Therefore, the invention of alternative technologies allowing early CRC and AA detection with minimal invasion is urgently needed.

Extracellular vesicles are various-sized membrane particles derived from host cells, which become a new source of biomarkers in liquid biopsy (*Shah et al., 2018*; *Min et al., 2021*). Small extracellular vesicles (sEVs) of 40–100 nm diameter isolated by 100 K ultracentrifugation were mostly considered to be exosomes, which derived from endosomes and participated in cell-cell communication (*An et al., 2015*). There are plenty of RNA species stuffed in sEVs, and one major function of sEVs is the delivery of functional donor cell RNAs to the recipient cell (*An et al., 2015*; *Thind and Wilson, 2016*). Additionally, the phospholipid bilayer of sEVs could effectively protect enclosed RNA from the RNase in the environment (*Théry, 2015*; *Vlaeminck-Guillem, 2018*), thus making sEV-RNA a relatively stable detection target. Consequently, for the diversity of enclosed RNAs and the properties of being protected from degradation by RNase (*Jeppesen et al., 2019*), the sEV-RNAs become an attractive treasury of biomarkers in cancer diagnosis.

Many studies have already reported the function of sEV-RNAs in the progress of many diseases and their potential application in diagnosis and prognosis (*Min et al., 2019a*; *Dorayappan et al., 2019*; *Ko et al., 2018*; *Cheng et al., 2018*). However, even though it is generally accepted that a landscape profiling of the contents from plasma EVs would largely facilitate the progress of biomarker discovery in liquid-biopsy of cancers, systematic screening of plasma sEV-RNAs is extremely hampered by the highly instrument-dependent, time-consuming isolation of sEVs from plasma and the exquisite manipulating of tiny amount RNA (*Srinivasan et al., 2019*). Previously, with a modified differential centrifugation (DC) procedure (*Wei et al., 2020*), we isolated sEVs from plasma and characterized those sEVs according to the MISEV2018 guideline (*Théry et al., 2018*). Then we proved that sEVs encapsuled miRNAs outperformed their free-floating counterparts in diagnostic liquid-biopsy, which proposed a new promising biomarker category (*Min et al., 2019b*). We also verified the abnormal abundance of several miRNAs in the circulating sEVs of CRC patients (*Min et al., 2019a*; *Min et al., 2019b*). However, even though the sEV miRNAs have already been carefully investigated, the panorama of circulating sEV-RNAs in CRC patients has not been revealed yet. Additionally, detecting AA by liquid biopsy could also be much more challenging than detecting CRC since AA always exhibited a smaller size and fewer vascular inside networks as compared to CRC (*Hong et al., 2018*).

To strengthen the ability of sEV biomarkers in identifying both T1a stage CRC and AA, here we accomplished the first whole-transcriptomic profiling of circulating sEV-RNAs in a large cohort with

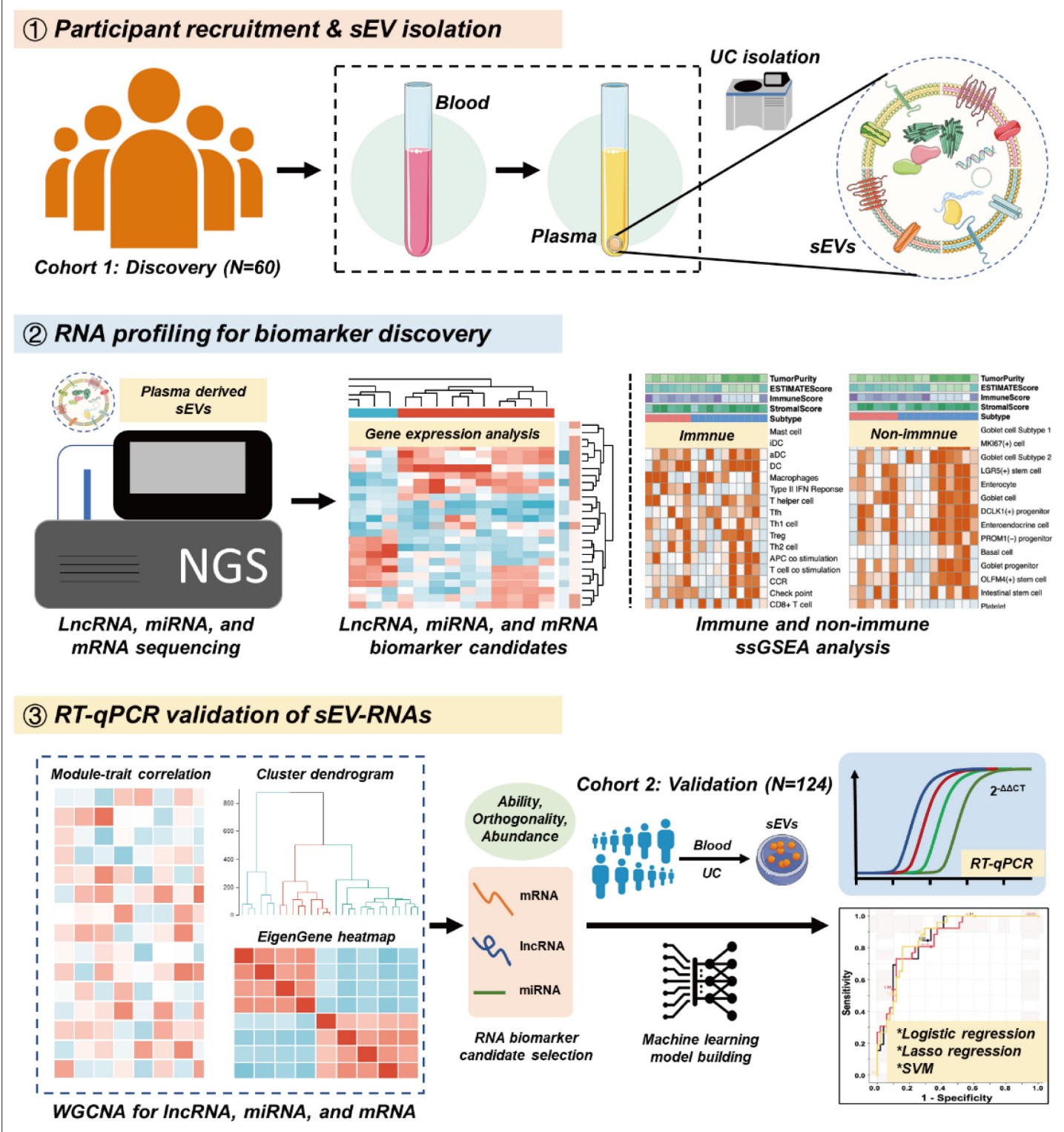

**Figure 1.** Schematic overview of the study design.

60 participants, including T1a stage CRC, AA, and normal controls (NC). The identification of CRC or AA-specific lncRNA and mRNA largely extended the sEVs biomarker repertoire, considering there were more than 50,000 lncRNA and mRNA species, nearly 25 times of known miRNA species (*Palazzo and Lee, 2015*). Finally, with a comprehensive analysis of plasma sEVs, we obtained the T1a stage

CRC and precancerous AA-specific sEV-RNA landscape, proposed a 10-gene signature, and verified its ability in identifying T1a stage CRC and AA in another cohort of 124 participants (*Figure 1*).

## Results

### Plasma-derived sEV characterization and RNA profiling

TEM imaging showed that the sEVs isolated from plasma exhibited cup-shaped, vesicle-like structures (*Figure 2a*). NTA analysis revealed a heterogeneous size distribution ranging between 75nm to 200nm (*Figure 2b*). Western blot analysis exhibited an enrichment of sEVs markers (CD9, TSG101, and Alix) and an absence of Calnexin, a negative marker of sEVs, indicating that the isolated fractions consisted mostly of sEVs (*Figure 2c*). The total RNA levels of the sEVs fractions were also assessed, and an EV-associated RNA concentration of 3.70±2.39 ng per mL plasma was reported.

For RNA profiling, a total of 58,333 annotated genes, including 2694 miRNAs, 24,927 mRNAs, and 30,712 lncRNAs, were detected. The numbers of detected miRNA species are almost equal among different groups, while there were slightly more mRNA and lncRNA species in T1a stage CRC than in NC/AA (*Figure 2d*). Differentially expressed genes (DEGs) among different groups were filtered by an FDR-adjusted ANOVA p-value. The top 100 miRNAs, mRNAs, and lncRNAs could roughly distinguish T1a stage CRC patients from AA and NC participants by unsupervised hierarchical clustering, respectively (*Figure 2e*). The clustering results using all those 300 RNAs exhibited clear discrimination between T1a stage CRC/AA patients and NC participants, and crude discrimination between T1a stage CRC and AA patients (*Figure 2—figure supplement 1*). Using the top 200 RNAs identified in *Figure 2e* based on p-values from the ANOVA algorithm for unsupervised t-SNE clustering provides a very clear separation of T1a stage colorectal cancer, advanced adenoma, and normal control individuals (*Figure 2f*, *Figure 2—figure supplement 2*).

The DEGs under comparisons between any two groups were identified by the Mann-Whitney U tests, and the overlaps of those DEGs were shown in a Venn diagram (*Figure 2g*, *Supplementary file 1-9*). Noticeably, there were 888 DEGs between AA and NC, and 519 (58%) of them overlapped with DEGs between T1a stage CRC and NC, suggesting most sEV-RNAs in AA remained abnormally expressed while progressing to T1a stage CRC. KEGG pathway analysis suggested that those DEGs were enriched in Pathways in cancer, MAPK signaling, Focal Adhesion, *etc.*, indicating a close relationship with cancer progression (*Figure 2h*). Additionally, the potential core regulatory networks between miRNAs and mRNAs in those DEGs were also exhibited (*Figure 2i–k*).

### Cell-specific features of the sEV-RNA profile indicated the different proportion of cells of sEV origin among different groups

We employed the ssGSEA algorithm to calculate the correlations between the sEV-RNA profile of each sample and the transcriptome characteristics of different cells to investigate the possible difference in the proportion of cells of sEV origin among different groups. The immune cell-specific features of each sample were shown in *Figure 3a*, and the stromal-related features were shown in *Figure 3b*. The complete landscape of cell-specific features of the sEV-RNA profile were shown in *Figure 3—figure supplement 1a*. Generally, plasma sEV-RNAs of different groups showed distinct cell-specific features. Especially, sEV-RNA of CC and RC showed higher expression levels of RNA markers of inflammation-promoting cells, MHC class I, HLA, TIL, and Paneth cells (*Figure 3c*). Correspondingly, RNA markers of DC cells, Mast cells, and Enterocyte cells were mostly enriched in sEV-RNAs of NC and AA compared to those of CC and RC (*Figure 3d*).

According to those cell-specific features, the tumor microenvironment-associated scores were calculated. The sEV-RNAs of CC, RC, and AA exhibited significantly higher Immune scores and significantly higher Estimate scores as compared to those of NC, whereas the NC group possessed the highest Stromal score (*Figure 3e*). Then we estimated the possible correlation among different cell-specific features, and a strong correlation among Inflammation promoting cells, MHC class I, HLA, TILs, Paneth cells, and Paneth-like cells was identified (*Figure 3f*, *Figure 3—figure supplement 1b*). Taking all together, cell-specific features of the sEV-RNA profile indicated that there is a higher proportion of inflammatory cell-originated sEVs in the plasma of CC, RC, and AA patients as compared to NC participants.

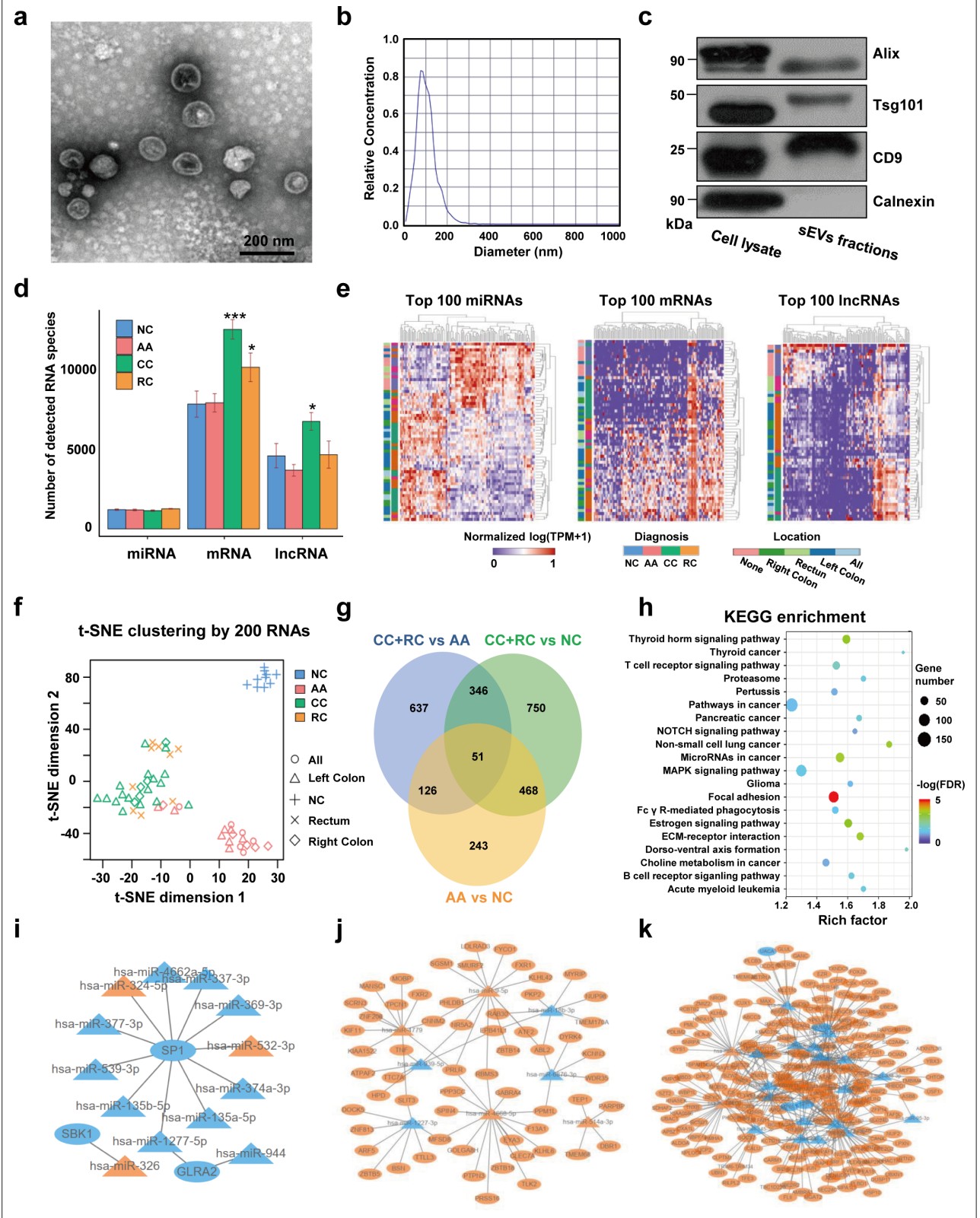

**Figure 2.** Transcriptome profiling of circulating sEVs. (**a**) TEM images of circulating sEVs isolated from human plasma. (**b**) NTA results of circulating sEVs enriched from plasma. (**c**) WB results of sEV positive (Alix, TSG101, CD9) and negative (Calnexin) markers. (**d**) The numbers of detected RNA species in different groups. (**e**) The hierarchical clustering results of top 100 miRNAs (left panel), mRNAs (middle panel), and lncRNAs (right panel). (**f**) t-SNE clustering by those candidate RNAs. (**g**) A Venn diagram showed DEGs shared between different comparisons (CRC vs NC, AA vs NC, CRC vs AA).

*Figure 2 continued on next page*

*Figure 2 continued*

(**h**) KEGG enrichment of all those DEGs identified. (**i–k**) potential core regulatory networks between miRNAs and mRNAs in DEGs identified in three (**i**), two (**j**), and one (**k**) of all comparisons.

The online version of this article includes the following source data and figure supplement(s) for figure 2:

**Source data 1.** Original file for the western blot analysis in *Figure 2C* (anti-Alix, anti-CD9, anti-TSG101, and anti-Calnexin).

**Source data 2.** Figures containing *Figure 2C* and original scans of the relevant western blot analysis (anti-Alix, anti-CD9, anti-TSG101, and anti-Calnexin) with highlighted bands and sample labels.

**Figure supplement 1.** The hierarchical clustering results of Top 100 miRNAs/mRNAs/lncRNAs.

**Figure supplement 2.** Unsupervised t-SNE clustering by those 200 RNAs with nine repeats.

## sEV-derived DEGs could be divided into six WGCNA modules

We performed WGCNA to reveal the inner correlation between DEGs and clinical parameters. Six coexpression modules were constructed based on the expression levels of 1525 DEGs in each sample by WGCNA (*Figure 4a*). The associations among different modules were shown in a heatmap (all DEGs with kME >0.7 in each module were used for calculating the Pearson correlation), revealing relatively high correlations between the green and turquoise modules, and between the blue and brown modules (*Figure 4b*). Additionally, the number and percentage of different RNA species in each module were also displayed in *Figure 4c* and *Figure 4d*, while the distributions of DEGs with kME >0.7 were shown in *Figure 4—figure supplement 1*. Noticeably, most miRNAs were categorized into the brown module, suggesting that the abundances of those sEV-miRNAs could be affected by the same factors, which further enhanced the necessity of including mRNAs and lncRNAs as additional biomarkers to cover as many patients as possible.

The expression levels of the top 10 DEGs in each module were shown in a heatmap (*Figure 4e*). We further used those DEGs for t-SNE analysis to cluster different samples and obtained a nearly perfect separation of T1a stage CRC, AA, and NC individuals (*Figure 4f*). In light of this result, we then tried to reduce the number of DEGs used for t-SNE analysis by picking only the top 5 DEGs and the top 1 DEG from each module, respectively. As shown in *Figure 4g*, when enrolled more DEGs of each module, the overall distinguishing effect significantly increased with less sample mis-clustered. Noticeably, the top 1 DEG from each module could separate the NC samples from T1a stage CRC and AA samples very clearly, meanwhile, CRC samples could also be blurrily separated from AA samples with only three exceptions. (*Figure 4h*).

## Different modules showed different expression trends

We conducted a GSEA analysis to extract major trends in the DEG expression of different modules. In the comparison between T1a stage CRC and NC, red, green, and turquoise DEGs were enriched in CRC samples, while black, blue, and brown DEGs were enriched in NC samples (*Figure 5a*). In the comparison between AA and NC, black, red, green, and turquoise DEGs were enriched in CRC samples, while blue and brown DEGs were enriched in NC samples (*Figure 5b*). In the comparison between T1a stage CRC and AA, black, brown, red, blue, and turquoise DEGs were enriched in CRC samples, while green DEGs were enriched in AA samples (*Figure 5c*).

The expression trends of the Top 10 DEGs of each module among NC, AA, and T1a stage CRC were also displayed (*Figure 5d–i*). Generally, green, red, turquoise and black DEGs are all overexpressed in AA as compared to NC. However, the expression level of green DEGs remained unchanged (*Figure 5d*), red/turquoise DEGs further increased to a much higher level (*Figure 5e,f*), while black DEGs almost decreased to the baseline level in CRC (*Figure 5g*). Blue and brown DEGs showed similar expression trends, which decreased to the lowest level in AA, and were partially restored in T1a stage CRC samples (*Figure 5h,i*).

## sEV-RNAs in different modules were correlated with different clinical factors

Detecting sEV-RNAs could also provide more data beyond diagnosis. Here, we performed module-trait correlation analysis to reveal clinical factors associated with those sEV-RNAs. The red module was significantly associated with endoscopic classification (*Figure 6a*). The black module was significantly

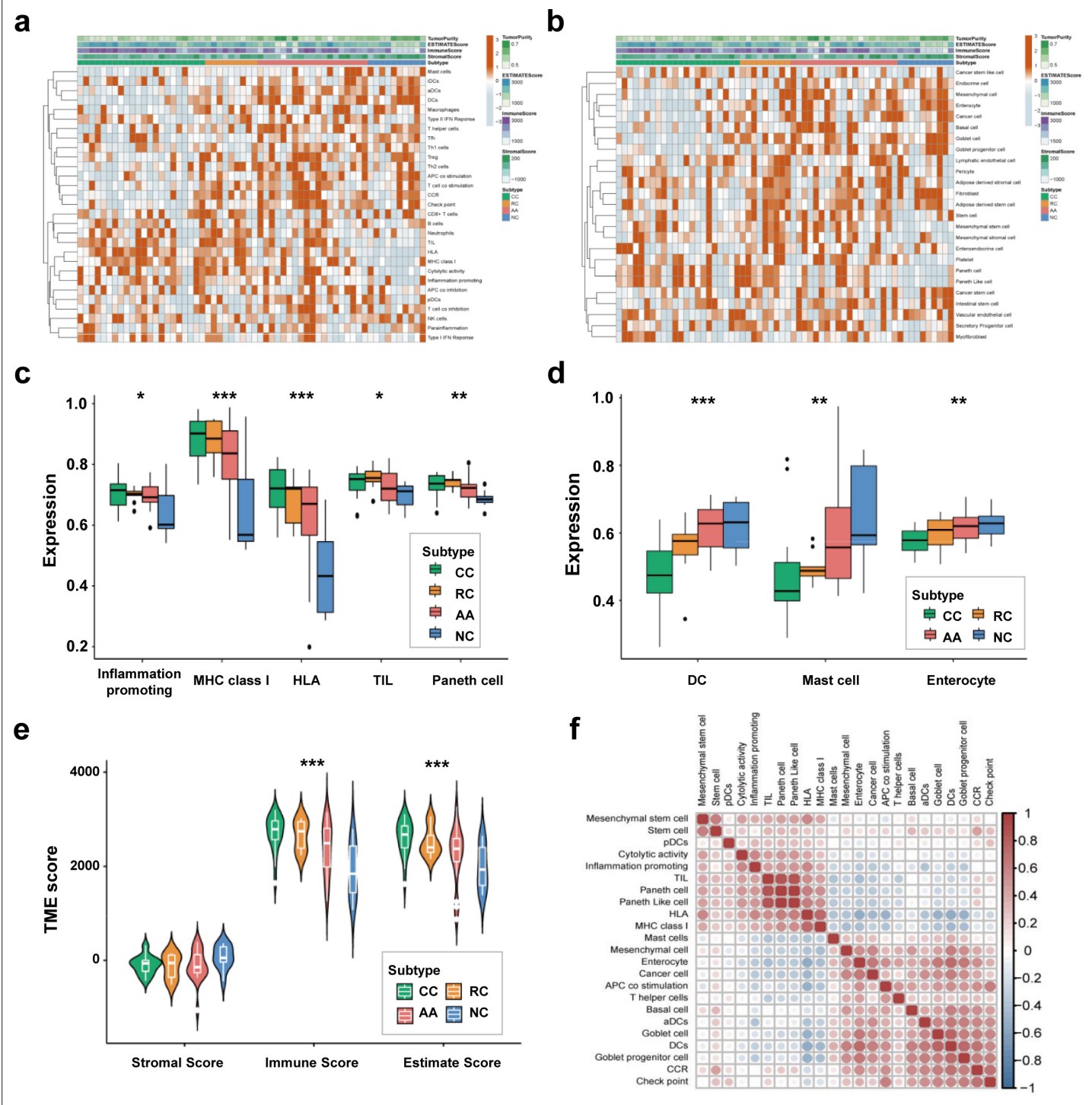

**Figure 3.** Cell-specific features of the sEV-RNA profile. (**a**) The hierarchical clustering heatmap of immune cell-specific features of each sample. (**b**) The hierarchical clustering heatmap of stromal-related features of each sample. (**c**) Boxplot of cell-specific features overexpressed in CC and RC patients (*p<0.05, **p<0.01, ***p<0.001). (**d**) Boxplot of cell-specific features overexpressed in NC participants (*p<0.05, **p<0.01, ***p<0.001). (**e**) The violinplot of the microenvironmental scores in different subgroups (*p<0.05, **p<0.01, ***p<0.001).(f) Correlation among cell-specific features differentially enriched among different groups.

The online version of this article includes the following figure supplement(s) for figure 3:

**Figure supplement 1.** Cell-specific features of the sEV-RNA profile.

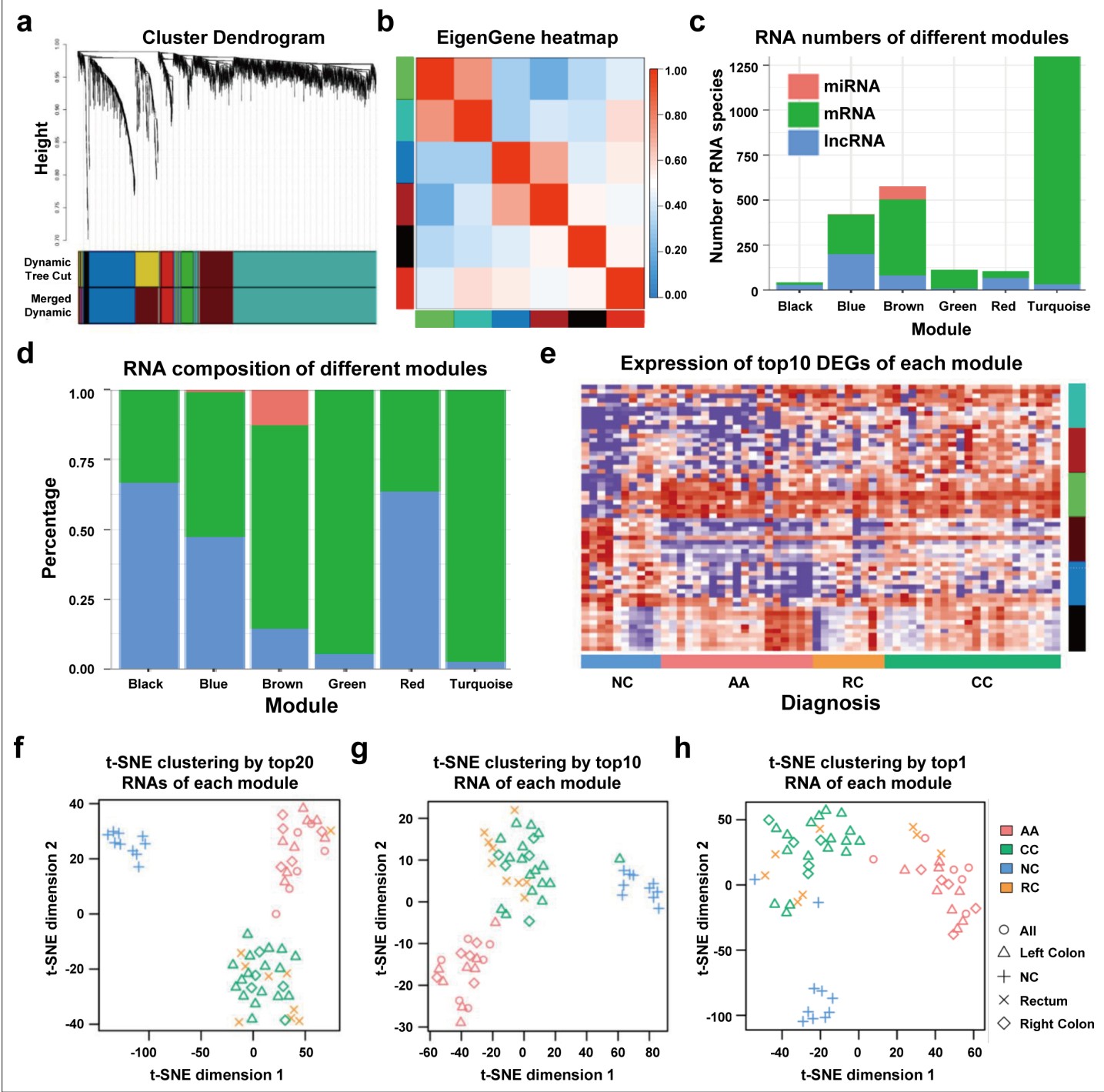

**Figure 4.** WGCNA analysis of sEV-RNAs. (**a**) Gene coexpression module construction of all DEGs identified in sEV-RNAs. (**b**) The heatmap exhibited Pearson correlations among different modules. (**c**) Bar plot of module composition of different modules (all DEGs). (**d**) Percentage bar plot of the RNA composition of different modules (all DEGs). (**e**) A heatmap exhibited the expression levels of the top 10 DEGs in each module. (**f**) t-SNE clustering by the top 10 DEGs in each module. (**g**) t-SNE clustering by the top 5 DEGs in each module. (**h**) t-SNE clustering by the top1 DEGs in each module.

The online version of this article includes the following figure supplement(s) for figure 4:

**Figure supplement 1.** Proportions and numbers of RNA species in different modules.

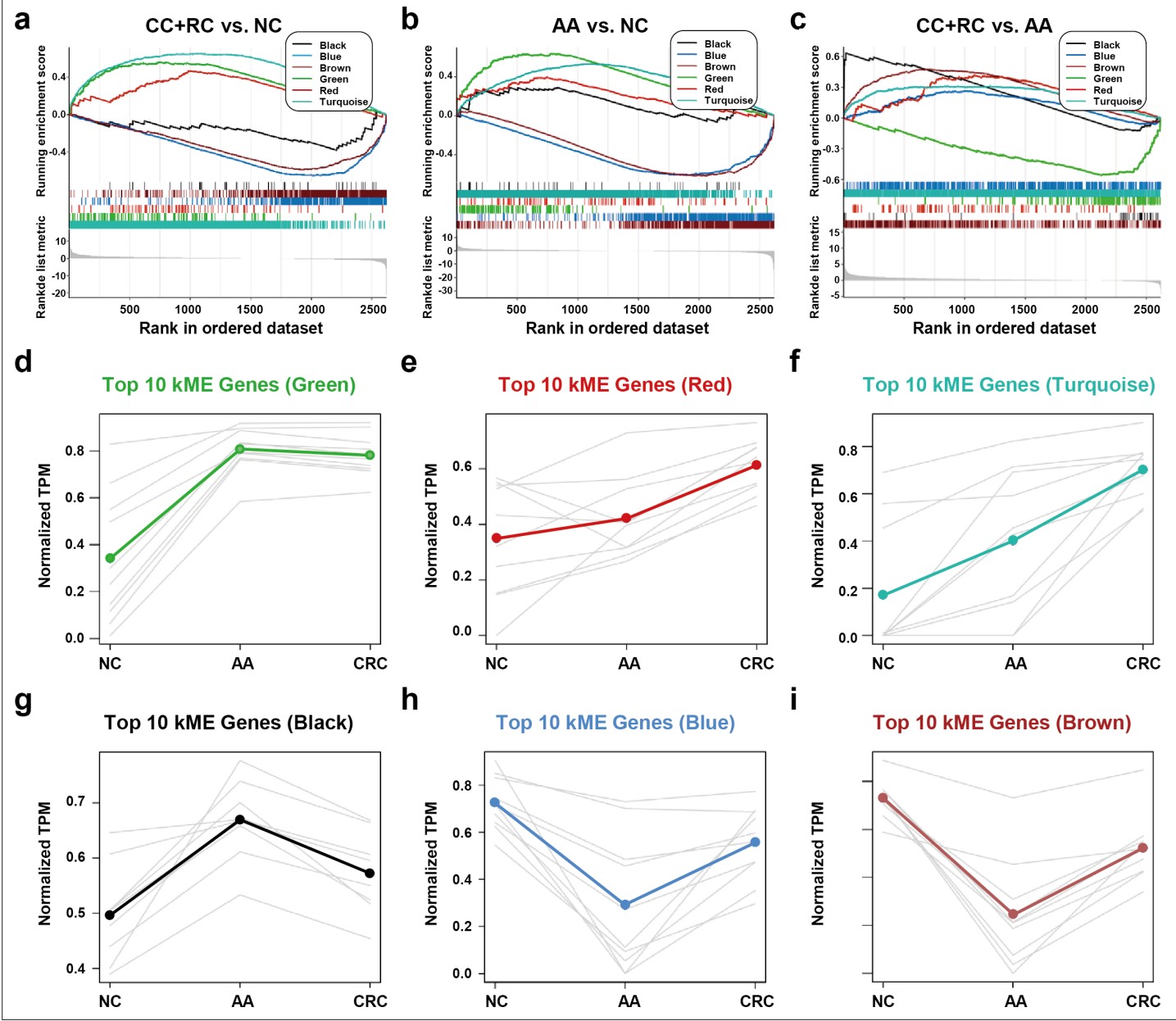

**Figure 5.** The expression trends of sEV-RNA modules. (**a–c**) GSEA analysis of DEGs in different modules (a: CRC vs. NC; b: AA vs. NC; c: CRC vs. AA). (**d-i**) The expression trends of the Top 10 DEGs of each module among NC, AA, and CRC (d: green module; e: red module; f: turquoise module; g: black module; h: blue module; i: brown module).

associated with the endoscopic classification and the LST morphology, while the green module was significantly associated with the LST morphology and the multiple-sited features (*Figure 6a*).

In brief, the DEG modules we identified in WGCNA showed a close association with morphological features of colorectal tumors, which was also well verified in RT-qPCR of representative sEV-RNAs. For example, flat lesions (subtype II) cases exhibited overexpressed MT-ND2 (blue module) levels in plasma sEVs as compared to protruding lesions (subtype Is/Ip/Isp) (*Figure 6b*, left panel), while LST tumors showed a relatively higher level of HIST2H2AA4 (green module) than non-LST tumors (*Figure 6b*, right panel).

Unsupervised hierarchical clustering suggested that sEV-RNAs of those three modules could not only separate T1a stage CRC/AA from NC but also roughly distinguish CRC and AA samples with different endoscopic classifications and other clinical features, such as LST and location (*Figure 6c*).

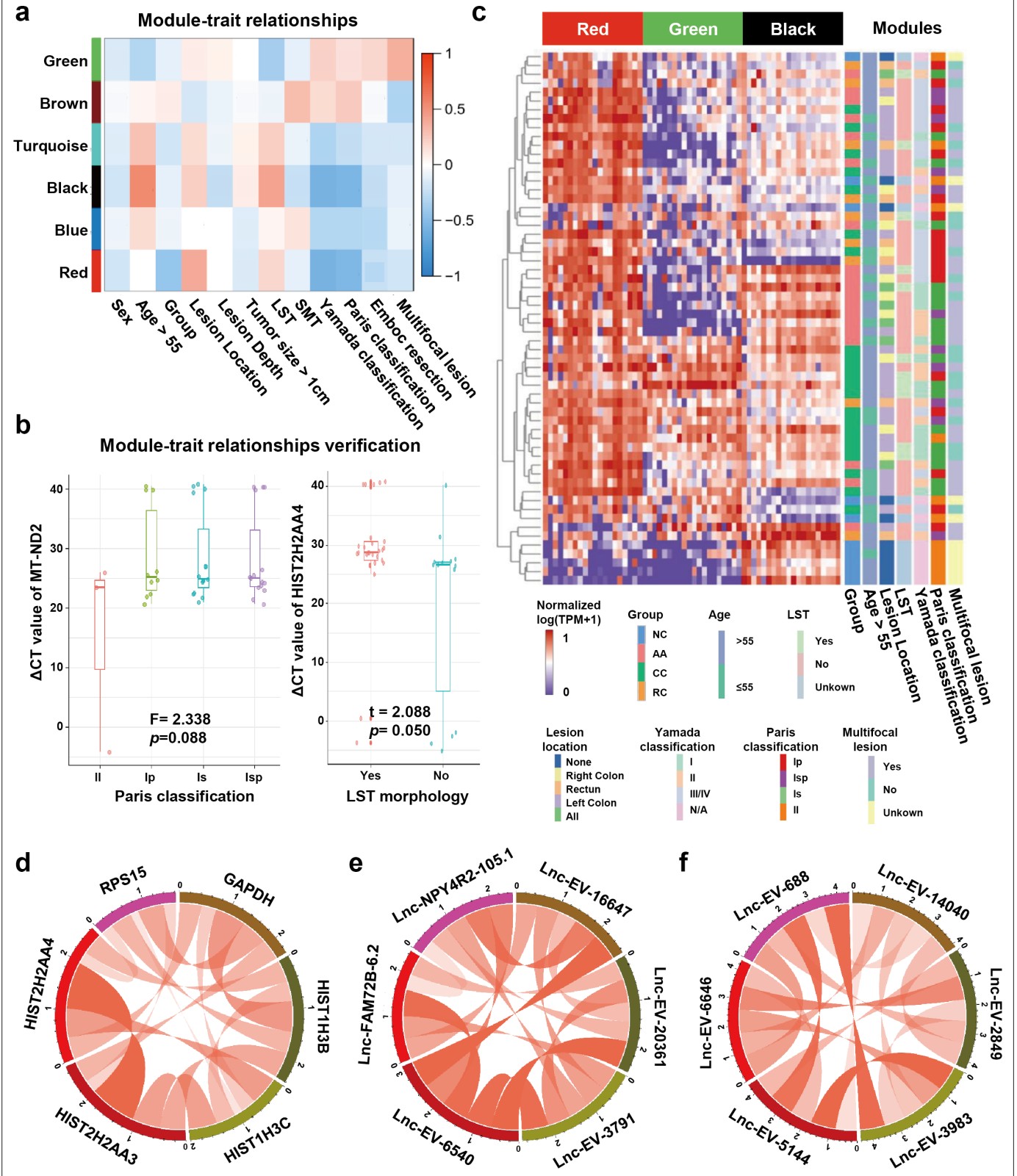

**Figure 6.** Module-trait correlation analysis of sEV-RNA modules. (**a**) The heatmap exhibited the correlation between modules and clinical traits. (**b**) The RT-qPCR validation of representative module-trait correlation (left panel: correlation between MT-ND2 and Paris classification; right panel: correlation between HIST2H2AA4 and LST morphology). (**c**) The heatmap exhibited the sEV-RNA expression levels of red, black, and green modules. (**d-f**) Circos plot showed the inner correlations among sEV-RNAs in the module green (**d**), red (**e**), and black (**f**).

The inner correlations among sEV-RNAs in the same module were also evaluated and exhibited by Circos plots. The top 2 sEV-RNAs of the green module exhibited a very strong correlation with each other (*Figure 6d*), while in other modules (red and black) the top 6 sEV-RNAs showed relative average correlation coefficients with each other (*Figure 6e,f*).

## Establishment of a sEV-RNA signature for AA and T1a stage CRC diagnosis

Even though the RNA signatures we established in *Figure 4f–h* showed promising performance in distinguishing T1a stage CRC, AA from NC, this next-generation sequencing (NGS)-based quantification system is not affordable in prospective large-scale screening of CRC. RT-qPCR is a cost-effective

**Table 1.** Selection of plasma sEVs-RNA candidates for AA and T1a stage CRC diagnosis.

| Candidate | AA vs NC | CRC vs NC | CRC vs AA | Module attribution | RNA Type | Amount | Finally seclected |
|---|---|---|---|---|---|---|---|
| miR-3615 | +* | + | -† | brown | miRNA | High | Yes |
| miR-330–5 p | + | + | - | brown | miRNA | Low | No |
| miR-425–5 p | + | + | - | NA | miRNA | High | Yes |
| miR-106b-3p | + | + | - | NA | miRNA | High | Yes |
| miR-589–5 p | + | + | - | NA | miRNA | Low | No |
| miR-181a-2–3 p | + | + | - | brown | miRNA | Low | No |
| Let-7f-5p | - | - | + | NA | miRNA | High | Yes |
| Let-7e-5p | - | - | + | NA | miRNA | High | No |
| miR-320a/b-3p | - | - | + | brown | miRNA | High | Yes |
| miR-664a-5p | - | - | + | brown | miRNA | Low | No |
| YBX3 | - | + | + | turquoise | mRNA | Low | No |
| C19orf43 | - | + | + | turquoise | mRNA | Medium | Yes |
| TOP1 | + | + | - | turquoise | mRNA | Medium | Yes |
| PPDPF | + | + | - | brown | mRNA | Medium | Yes |
| MT-ND2 | + | + | - | blue | mRNA | High | Yes |
| HIST2H2AA4 | + | + | - | green | mRNA | Medium | Yes |
| RPL10 | + | + | - | green | mRNA | High | No |
| RPS29 | + | + | - | blue | mRNA | High | No |
| IST1 | - | + | + | black | mRNA | Low | No |
| CSE1L | - | + | - | red | mRNA | Low | No |
| lnc-MSI1-2:1 | + | + | - | brown | lncRNA | High | Yes |
| lnc-FCGR1B-16:1 | - | + | + | red | lncRNA | Low | No |
| lnc-NPY4R2-105:1 | - | + | + | red | lncRNA | Medium | No |
| lnc-MKRN2-42:1 | - | + | + | turquoise | lncRNA | High | Yes |
| LNC_EV_9572(Chr8: 34358093–34456247) | + | + | - | black | lncRNA | High | Yes |
| LNC_EV_21004(Chr21: 8212554–8440060) | + | + | - | brown | lncRNA | High | No |
| LNC_EV_15260(Chr14: 49555875–49923916) | + | + | - | turquoise | lncRNA | High | No |

*+: significant difference found in this comparision.
†-: no significant difference found in this comparision.

**Table 2.** CRC and AA prediction models established by Lasso regression.

Additionally, we adopted quadratic discriminant analysis (QDA) to demonstrate the possibility of the plasma sEV-RNA signature for direct sample classification. An overall accuracy of 78% (ranging from 63% to 100%) was obtained for direct sample classification (*Figure 7ij*). Generally, the individuals classified into AA/CC/RC are considered as high-risk and should be advised to further endoscopic examination, and our QDA classifier provided a specificity of 79.25%, and a sensitivity of 99.0% (with only one CC sample missed) in identifying those high-risk individuals. Together, our RT-qPCR-based plasma sEVs-RNA signature could be a powerful and better alternative to FIT and FOBT tests in CRC and precancerous AA screening programs.

| Model type | Signature | RNAs | Lambda | AUC |
|---|---|---|---|---|
| CRC prediction | 5-RNA signature | Let-7f-5p, C19orf43, TOP1, PPDPF, lnc-MKRN2-42:1 | 0.05 | 0.73 |
| | 6-RNA signature | Let-7f-5p, C19orf43, TOP1, PPDPF, lnc-MKRN2-42:1, LNC-EV-9572 | 0.035 | 0.73 |
| | 7-RNA signature | Let-7f-5p, C19orf43, TOP1, PPDPF, lnc-MKRN2-42:1, LNC-EV-9572, HIST2H2AA4 | 0.03 | 0.74 |
| | 8-RNA signature | Let-7f-5p, C19orf43, TOP1, PPDPF, lnc-MKRN2-42:1, LNC-EV-9572, HIST2H2AA4, miR-320a-3p | 0.02 | 0.76 |
| AA prediction | 6-RNA signature | miR-425–5 p, Let-7f-5p, C19orf43, TOP1, PPDPF, LNC-EV-9572 | 0.05 | 0.83 |
| | 7-RNA signature | miR-425–5 p, Let-7f-5p, C19orf43, TOP1, PPDPF, LNC-EV-9572, lnc-MKRN2-42:1 | 0.04 | 0.84 |
| | 8-RNA signature | miR-425–5 p, Let-7f-5p, C19orf43, TOP1, PPDPF, LNC-EV-9572, lnc-MKRN2-42:1, HIST2H2AA4 | 0.1 | 0.87 |
| | 9-RNA signature | miR-425–5 p, Let-7f-5p, C19orf43, TOP1, PPDPF, LNC-EV-9572, lnc-MKRN2-42:1, HIST2H2AA4, MT-ND2 | 0.05 | 0.88 |

alternative to NGS in diagnostic tests, thus here we designed an RT-qPCR-based assay to diagnose CRC and AA patients.

We extracted all sEV-RNAs upregulated in CRC or AA using a 4-fold change cutoff (*Supplementary file 10 and 11*). High-abundance sEV-RNAs were considered more appropriate for RT-qPCR quantification since they are much easier to detect in liquid biopsy-obtained trace samples. Therefore, we excluded all low-abundance sEV-RNAs (median TPM <50). A signature covering more different modules is preferred since it includes more orthogonal factors in prediction. Thus, the candidate sEV-RNAs were finally selected based on their fold change in CRC/AA, absolute abundance, and module attribution (*Table 1*).

## Simultaneously detecting AA and CRC by a RT-qPCR-based sEV-RNA signature

The expression levels of those candidate sEV-RNAs were quantified by RT-qPCR in an independent cohort of 124 participants. Lasso regression was applied to select the most effective variables from all candidate sEV-RNAs to construct a multivariate CRC prediction model (*Figure 7—figure supplement 1*). The model with 8 sEV-RNAs exhibited an AUC of 0.76 (*Table 2*: upper panel; *Figure 7a–d*). Three different algorithms (logistic regression, Lasso regression, and SVM) were compared to further boost the performance of the 8-gene signature, and the highest AUC of 0.80 was achieved in the SVM model (*Figure 7a–d*). In predicting Stage I CRC patients, these sEV-RNAs panels also exhibit a promising predictive performance (*Figure 7—figure supplement 2*). Furthrtmore, we assessed the discriminative effects of CRC on NC, taking into account different age groups, genders, tumor sizes, and tumor anatomical locations. To minimize the potential overfitting effect due to the reduction in sample size after partitioning, we implemented a 10-fold cross-validation for each panel and these sEV-RNAs panels exhibit promising performance in different clinical parameters (*Figure 7—figure supplement 3*).

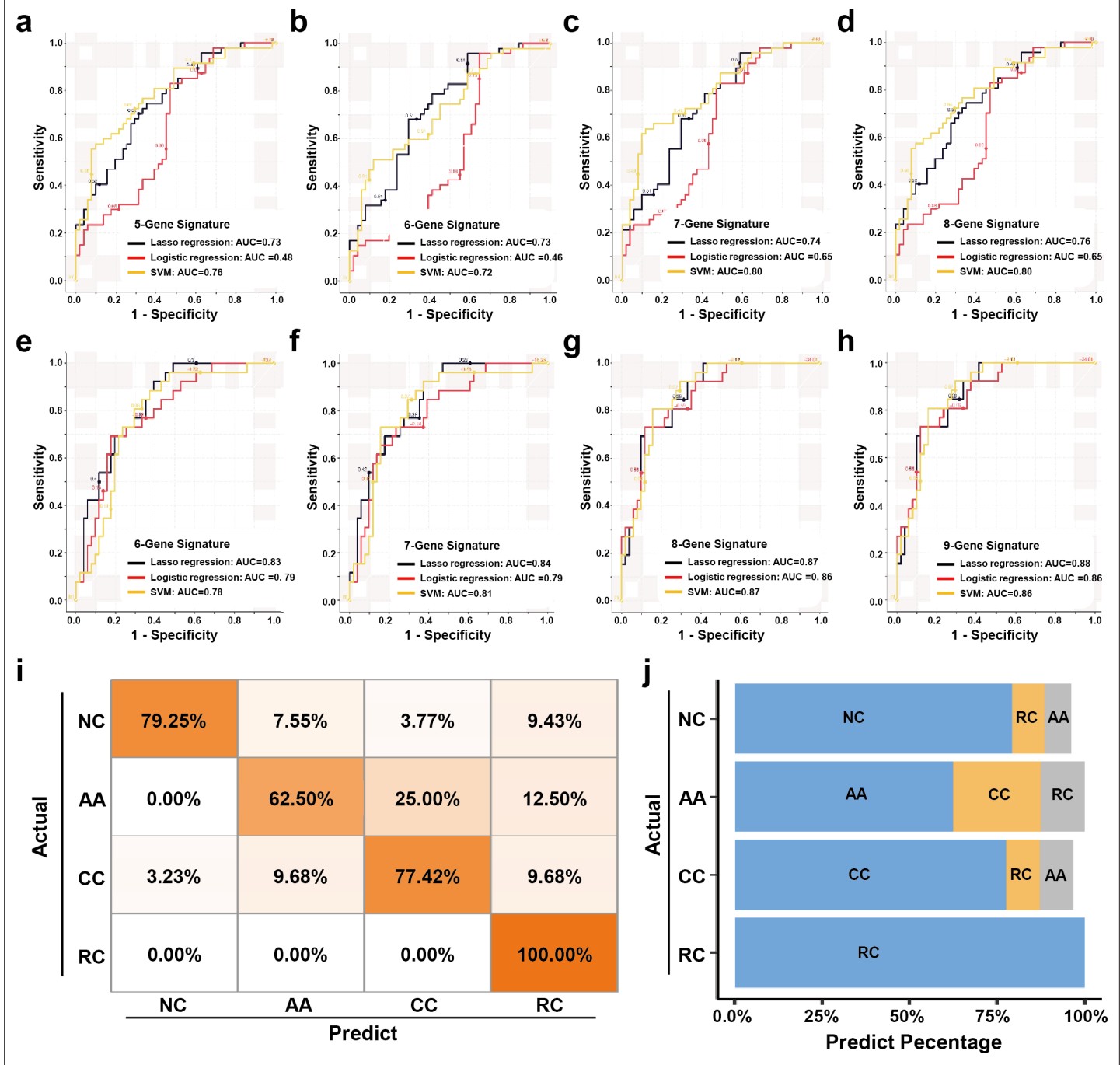

**Figure 7.** The plasma sEVs-RNA signature to detect early CRC and AA. (**a–d**) The ROC analysis of different sEV-RNA signatures in the prediction of CRC patients by different algorithms (a: 5-gene panel; b: 6-gene panel; c: 7-gene panel; d: 8-gene panel). (**e–h**) The ROC analysis of different sEV-RNA signatures in the prediction of AA patients by different algorithms (e: 6-gene panel; f: 7-gene panel; g: 8-gene panel; h: 9-gene panel). (**i**) The QDA results of all 13 sEV-RNAs in classifying all samples. (**j**) Statistical summary of QDA performance in each sample group.

The online version of this article includes the following figure supplement(s) for figure 7:

**Figure supplement 1.** Lasso regression to construct multivariate prediction models.

**Figure supplement 2.** The ROC analysis of different sEV-RNA signatures in the prediction of stage I CRC patients by different algorithms (**a**: 6-gene panel; **b**: 7-gene panel; **c**: 8-gene panel; **d**: 9-gene panel).

**Figure supplement 3.** The ROC analysis of different sEV-RNA signatures for predicting CRC patients using the Lasso regression algorithm in different clinical parameters (ab: age; cd: gender; ef: tumor size; gh: anatomical position).

Similarly, a model with 9 sEV-RNAs exhibited an AUC of 0.88 to predict AA from NC (*Table 2*: lower panel; *Figure 7e–h*). The Lasso model showed the highest AUC of 0.88 among all three algorithms, signifying an excellent efficacy in AA detection among noncancerous individuals (*Figure 7e–h*).

## Discussion

Previously, we proved that plasma sEV-encapsulated miRNAs outperformed their free-floating counterparts in the diagnostic liquid biopsy of cancer (*Min et al., 2019b*). However, whole-transcriptomic profiling of plasma sEVs is much more challenging and highly attractive, as there are many more long transcript species than miRNAs in plasma sEVs (*Jeppesen et al., 2019*). Additionally, mRNAs and lncRNAs encapsulated in sEVs manifested very important functions during carcinogenesis and cancer progression. For example, lncARSR in sEVs promoted sunitinib resistance in renal cancer in a ceRNA-dependent manner (*Qu et al., 2016*). Similarly, sEV-delivered HChrR6 mRNA showed a therapeutic effect in the treatment of human HER2(+) breast cancer (*Forterre et al., 2020*). Thus, whole-transcriptomic profiling of circulating sEVs would not only facilitate the minimally invasive diagnosis of T1a stage CRC but also promote our understanding of the microenvironment change during CRC carcinogenesis.

Here, we identified 2694 miRNAs (1838 known, 856 novel), 24,927 mRNAs (18,947 known, 5980 novel), and 30,712 lncRNAs (12,928 known, 17,784 novel) in human circulating sEVs. A total of 2621 RNA species showed a primary potential in distinguishing colorectal tumors from others, which was a great treasury of biomarker candidates in liquid biopsy. The profiling of circulating sEV-RNAs for early detection of CRC and AA we provided here, which is, to our knowledge, not only the first published whole-transcriptomic profile (including miRNA, mRNA, and lncRNA) dataset from human plasma EVs, but also the first attempt to simultaneously diagnose early CRC and precancerous AA by a plasma sEV-RNA signature. It is also worth noting that *Zheng et al., 2020* reported a data-independent acquisition (DIA)-mass spectrometry (MS)-based protein profiling of circulating sEVs. The plasma sEV-protein signature exhibited very good performance, especially for CRC patients with liver metastases. Thus, here we also suggested that a comprehensive biomarker panel consisting of both plasma sEV-RNAs and proteins could be very promising in identifying both early and advanced CRC patients in the future.

Our results proved that sEV-RNAs could well separate T1a stage CRC, AA, and NC from each other, indicating a significant overall change of plasma sEV-RNA profile during the whole process of CRC carcinogenesis. We also found that those sEV-RNAs were enriched in pathways associated with cancer, MAPK signaling, Focal adhesion, etc., suggesting those sEV-encapsulated RNAs could be potentially effective. Noticeably, those sEV-RNAs were not necessarily originated from cancer cells, since target-selecting procedures, such as anti-EpCAM-based immuno-capture (*Dorayappan et al., 2019*; *Ostenfeld et al., 2016*; *Zhang et al., 2019*), were not conducted before sEVs isolation. Nevertheless, biomarker studies that enrolled all possible biomarkers regardless of their origination could provide a larger candidate pool, a higher abundance and detectability, and a simplified procedure for easier detection of given targets. Detailed single sEV analysis or other methodology revealing sEV-RNA heterogeneity (*Colombo et al., 2013*; *Lee et al., 2018*) could further improve the performance of sEV-based liquid biopsy and reveal the underlying molecular mechanisms.

Here we provided a 6-RNA signature based on the RNA-seq data, which exhibited a good effect in distinguishing T1a stage CRC, AA from NC. The simplicity of this signature is quite inspiring but not surprising, considering it took advantage of the discrimination efficiency of sEV-RNAs and the orthogonality between WGCNA modules (*Qin et al., 2019*). Some but not all of the key sEV-RNAs we identified have already been reported involved in carcinogenesis. For example, miR-425–5 p was reported in promoting CRC via BRAF/RAS/MAPK Pathways (*Angius et al., 2019*). Hypoxia-induced let-7f-5p regulated osteosarcoma cell proliferation and invasion by inhibiting the Wnt signaling (*Chen et al., 2020*). Additionally, patients with higher PPDPF expression exhibited worse prognosis than others (*Mao et al., 2019*), while high HIST1H2BK levels predicted poor prognosis in glioma patients (*Liu et al., 2020*). Further investigation on the functions of other key sEV-RNAs was needed to provide a full landscape of the molecular mechanism of their roles during carcinogenesis.

To facilitate the clinical application of sEV-RNAs in CRC screening, an RT-qPCR-based assay was designed. RNA candidates were selected from the sEV-RNAs identified by RNA-seq under a tradeoff between expected performance and RT-qPCR detectability, and sEV-RNAs with higher abundance

distributed in different modules were preferred. Although we got a good performance for this sEV-RNA signature, other gene selection strategies were also worth trying and improved results could also be possibly achieved by a more meticulous pipeline considering other factors such as sample-specific effects (*Yu et al., 2020*). To maximize the performance of given sEV-RNA signatures, we compared the traditional logistic regression model with two popular machine learning algorithms, that is SVM and Lasso regression. The SVM model and Lasso model showed higher AUC in identifying T1a stage CRC and AA, respectively. QDA analysis was also tested for a direct classification of different samples, which exhibited a specificity of 79.25%, and a sensitivity of 99.0%. Those efforts also highlighted the power of modeling algorithm selection in biomarker discovery and utilization.

Detecting sEV-RNAs could also provide more data beyond diagnosis. The modules we identified in WGCNA showed a very close association with morphological features of colorectal tumors, which was also verified in RT-qPCR analysis of representative sEV-RNAs. For example, LST tumors showed a relatively higher level of plasma sEV-derived HIST2H2AA4 than non-LST tumors, while flat lesions exhibited overexpressed plasma sEV-derived MT-ND2 levels than protruding lesions. Considering the higher HIST2H2AA4 mRNAs and MT-ND2 mRNAs in tissues could reflect a higher proliferation rate and higher energy metabolism (*Liu et al., 2020*; *Sun et al., 2009*), their elevated levels in sEVs could be a potential indicator of invasiveness. Although those interpretations are hypothetical, the fact that one could get biological information on cancerous and precancerous tumors from plasma sEVs is still inspiring. Hence we believed that in-depth data mining of sEV-RNAs acquired by liquid biopsy would be helpful in different scenarios of healthcare decision-making.

## Conclusions

We demonstrated the clinical value of circulating sEV-RNA profiling in CRC biomarker discovery and established a high-accuracy, low-cost RNA signature that could detect both T1a stage CRC and AA from other individuals. We also suggested that T1a stage CRC and precancerous AA patients retained a specific plasma sEV-RNA profile, which would provide insight into an in-depth understanding of the mechanism of CRC carcinogenesis.

## Methods

### Patients' information and plasma collection

For biomarker discovery, 31 early (T1a stage) CRC patients, including 22 colon cancer (CC) patients and 9 rectum cancer (RC) patients who received endoscopic resection at Department of Gastroenterology, Beijing Friendship Hospital between January 2018 and December 2019 were enrolled along with 19 AA patients (characterized by high-grade dysplasia or adenomas ≥10 mm or ≥25% villous component) and 10 non-cancerous (NC) outpatients with other gastrointestinal symptoms. Additional 124 participants (47 CRC, 24 AA, and 53 NC) were enrolled for validation. All clinical information was summarized in *Supplementary file 12*. This study was approved by the ethics committee of Beijing Friendship Hospital, and written informed consent was obtained from each participant.

A total of 6 mL blood from each individual was collected in EDTA tubes in the morning before any food/water intake. After two-step centrifugation at 1300×*g*, 15 min (isolate plasma from whole blood) and 3000×*g* for 15 min (platelet removal) at 4 °C, the supernatant of the processed plasma was aspirated and stored at –80 °C before use.

### sEV isolation by a modified DC procedure

A total of 3 mL plasma was collected from each participant. The DC isolation procedure was performed according to our previous report (*Min et al., 2019b*). Briefly, the plasma was centrifuged at 3000×*g* for 15 min and 13,000×*g* for 30 min, to remove cell debris and large EVs respectively, and the final supernatant was 1:7 diluted by PBS, filtered by a 0.22 μm sieve, ultracentrifuged using a P50AT2-986 -rotor (CP100NX; Hitachi, Brea, CA) at 150,000×*g*, 4 °C for 4 h to collect the sEVs. The pellet was washing in PBS and centrifuged again at 150,000×*g*, 4 °C for 2 h. Then the sEVs enriched fraction was re-suspended in 100 μL PBS. The full description of methodologies was also submitted to EV-TRACK (ID: EV190033) (*Van Deun et al., 2017*).

### sEV characterization

The protein of sEVs was quantified by a BCA Assay Kit (Thermo Fisher Scientific, Product No. 23225) according to the manufacturer's protocol. A total of 10 μg protein was loaded to each lane for western

blot analysis. The detailed procedure was referred to our previous publication (*Min et al., 2019b*), using CD9 antibody (60232–1, Proteintech, Wuhan, China), TSG101 antibody (sc-136111, Santa Cruz, CA, USA), Alix antibody (sc-53540, Santa Cruz, CA, USA) and calnexin antibody (10427–2-AP, Proteintech, Wuhan, China).

Nanoparticle tracking analysis (NTA) and transmission electron microscopy (TEM) were used for morphological characterization, which was conducted according to our previous publication (*Min et al., 2019b*). For NTA, sEVs suspension (ranging from 1×10^7 /mL to 1×10^9 /mL) was detected by the ZetaView PMX 110 (Particle Metrix, Meerbusch, Germany) with a 405 nm laser to determine the size and quantity of sEVs. A 60 s video was taken with a 30 frames/s frame rate, and the movements of those nanoparticles were analyzed using ZetaView 8.02.28. For TEM, about 15 µL sEVs suspension was placed on a copper mesh and incubated at room temperature for 10 min. After washing with sterile distilled water, the sEVs solution was contrasted by the uranyl-acetate solution for 1 min. Then the sample was dried for 2 min and observed under a TEM microscope (JEOL-JEM1400, Tokyo, Japan).

## Data processing outline based on RNA sequencing

ExoRNA was isolated by the miRNeasy Mini kit (No. 217004, Qiagen, Hilden, Germany). RNA degradation and possible DNA contamination were monitored on a 1.5% agarose gel. Two different RNA sequencing libraries (long RNA, short RNA) were constructed, respectively. The detailed procedures for RNA library preparation and sequencing are outlined below. Furthermore, detailed information on the identification, quantification, and differential expression analysis of miRNAs, mRNAs, and lncRNAs has been provided. The RNA sequencing data have been deposited at the Sequence Read Archive (SRA) database of NCBI under the accession number PRJNA639943.

## RNA library preparation and sequencing

For long RNA libraries, a total amount of 5 ng RNA per sample was used as input material for sequencing libraries using the Ovation SoLo RNA-Seq Library Preparation Kit (NuGEN, CA, USA) following the manufacturer's recommendations and index codes were added to attribute sequences to each sample. For small RNA libraries, 2.5 ng RNA per sample was used as input material for the RNA sample preparation. Sequencing libraries were generated using NEB Next Multiplex Small RNA Library Prep Set for Illumina (NEB, USA) following the manufacturer's recommendations and index codes were added to attribute sequences to each sample.

Then PCR products were purified (AMPure XP system) and library quality was assessed on the Agilent Bioanalyzer 2100. The clustering of the index-coded samples was performed on a cBot Cluster Generation System using TruSeq PE Cluster Kitv3-cBot-HS (Illumia). After cluster generation, the library preparations were sequenced on an Illumina Hiseq platform and paired-end reads were generated.

## Identification, quantification, and differential expression analysis of miRNAs

The clean reads were aligned to the reference GRCh38 using Bowtie tools. Annotated by the Silva database, GtRNAdb database, Rfam database, and Repbase, ribosomal RNA (rRNA), transfer RNA (tRNA), small nuclear RNA (snRNA), small nucleolar RNA (snoRNA), other ncRNA and repeats were filtered before further analysis. The remaining reads were used to quantify known miRNA and predict new miRNA by compared to miRNAs from miRbase and GRCh38, respectively. Read count for each miRNA was obtained from the mapping results, and TPM was calculated. TMM normalization was performed and DEGs analysis of any two groups was conducted using the Mann Whitney U test with p-value <0.05, and Fold change >1.5.

## Identification of mRNAs and lncRNAs

Raw data (raw reads) of fastq format were firstly processed through in-house Perl scripts. In this step, clean data (clean reads) were obtained by removing reads containing adapter, reads containing ploy-N, and low-quality reads. At the same time, Q20, Q30, GC-content, and sequence duplication level of the clean data were calculated. All the downstream analyses were based on clean data with high quality. Paired-end clean reads were aligned to the reference GRCh38 using TopHat2/Bowtie2. Mapped reads were used for the quantification of mRNA level and differential expression analysis.

For lncRNA analysis, the transcriptome was assembled using the Cufflinks and Scripture based on the reads mapped to the reference genome. The assembled transcripts were annotated using the Cuffcompare program from the Cufflinks package. The unknown transcripts were used to screen for putative lncRNAs. Three computational approaches including CPC (Coding Potential Calculator)/CNCI (Coding-Non-Coding Index)/Pfam were combined to sort non-protein coding RNA candidates from putative protein-coding RNAs in the unknown transcripts. CPC is a sequence alignment-based tool used to assess protein-coding capacity. By aligning transcripts with known protein databases, CPC evaluates the biological sequence characteristics of each coding frame of the transcript to determine its coding potential and identify non-coding RNAs (*Kong et al., 2007*). CNCI analysis is a method used to distinguish between coding and non-coding transcripts based on adjacent nucleotide triplets. This tool does not rely on known annotation files and can effectively predict incomplete transcripts and antisense transcript pairs (*Sun et al., 2013*). Pfam divides protein domains into different protein families and establishes statistical models for the amino acid sequences of each family through protein sequence alignment (*Finn et al., 2014*). Transcripts that can be aligned are considered to have a certain protein domain, indicating coding potential, while transcripts without alignment results are potential lncRNAs. Putative protein-coding RNAs were filtered out using a minimum length and exon number threshold. Transcripts above 200 nt with more than two exons were selected as lncRNA candidates and further screened by CPC/CNCI/Pfam. We distinguished lncRNAs from protein-coding genes by intersecting the results of the three determination methods mentioned above. Considering the current limited understanding of various sEV-lncRNAs in research, further exploration is necessary in the future to better elucidate the roles of different RNA categories involved in the tumorigenesis process.

## Quantification and differential expression analysis of mRNAs and lncRNAs

Stringtie was used to calculate FPKMs of coding genes in each sample. Gene FPKMs were computed by summing the FPKMs of its all alternatively spliced transcripts. Genes with median FPKM <5 were regarded as low abundance genes and excluded in the subsequent analysis. TMM normalization was performed and differentially expressed genes (DEGs) analysis of any two groups was conducted using the Mann Whitney U test with cutoff *P*-value <0.05, and Fold change >1.5.

## Single sample gene set enrichment analysis (ssGSEA)

'GSVA' R package (version 1.42.0) was employed to analyze the cell-specific features of the sEV RNA profiles. One-way ANOVA test was employed to identify the differentially enriched cell-specific features among different groups. 'ESTIMATE' R package (version 1.0.13) was used to estimate the stromal score, immune score and microenvironmental score.

## Weighted gene coexpression network analysis (WGCNA)

The 'WGCNA' R package (version 1.61) was used to construct a co-expression network for all differentially expressed genes (DEGs). All samples were used to calculate Pearson's correlation matrices. The weighted adjacency matrix was created with the formula amn = |cmn|β (amn: adjacency between gene m and gene n; cmn: Pearson's correlation between gene m and gene n; β: soft-power threshold). Furthermore, the weighted adjacency matrix was transformed into a topological overlap measure (TOM) matrix to estimate its connectivity property in the network. Average linkage hierarchical clustering was used to construct a clustering dendrogram of the TOM matrix. The minimal gene module size was set to 30 to obtain appropriate modules, and the threshold to merge similar modules was set to 0.1.

## GeneSet enrichment analysis (GSEA) and t-SNE clustering

GSEA is supported by the 'DOSE' R package (version 3.8.0). Each comparison (CRC vs NC, AA vs NC, CRC vs AA) was used as a phenotype label, while the gene list of each module obtained by WGCNA was adopted as a gene set. A signal-to-noise metric was used for ranking genes and all other parameters are all set as default.

The 'Rtsne' R package (version 0.15) was used for t-distributed stochastic neighbor embedding (t-SNE) clustering. RNAseq data from all 60 samples were analyzed using t-SNE-based clustering of DEGs.

## Selection of predictive biomarkers for model inclusion

To ensure the predictive performance of the sEV-RNA signature, candidate sEV-RNAs were ultimately selected based on their fold change in colorectal cancer/precancerous advanced adenoma, absolute abundance, and module attribution. In detail, we initially selected the top 10 RNAs from each category (mRNA, miRNA, and lncRNA) with a fold change greater than 4. In cases where fewer than 10 RNAs were meeting this criterion, all RNAs with a fold change greater than 4 were included. Subsequently, we filtered out RNAs with low abundance, and we selected the top-ranked RNAs from each module based on the fold change ranking for inclusion in the final model.

## Quantification of RNA expression with qPCR

Synthetic *Caenorhabditis elegans* cel-39–3 p was spiked into each sEVs sample as an external calibration before RNA extraction. Total RNA was isolated and purified from sEVs factions by a miRNeasy Mini kit (cat. 217004, Qiagen, Hilden, Germany). The total RNA was then reversely transcribed to synthesize cDNA using the PrimeScript RT reagent Kit (TAKARA, RR037A). The abundance of target gene expression was detected by the TaqMan probe using real-time qPCR. Two μL of cDNA was used as the template for each PCR reaction. The sequences of primers and probes were shown as *Supplementary file 13*. All samples were normalized by the initial biofluid input volume used for RNA extraction and calibrated by the amount of spike-in cel-39–3 p.

## Statistical analysis

Statistical analysis were all performed by R 3.8.2 (https://www.r-project.org). $p < 0.05$ was considered significant, and FDR adjusted p-value was calculated for multiple comparisons. All tests were two-tailed. The efficiency of candidate RNA models was assessed by calculating the area under the ROC curve (AUC). Pheatmap, VennDiagram, and ggplot2 were used for data visualization.

## Acknowledgements

We thank Dr. Da Qin and Ms. Rui Wei (Capital Medical University) for their discussion and valuable advice on this study. We thank Ms. Yuan Wang, Ms. Jing Chen, Dr. Guanyi Kong (Echo Biotech Co., Ltd) for the meticulous work in project coordination. We thank Ms. Yun Zhang and the Clinical Data and Biobank Resource of Beijing Friendship Hospital for their help in sample preservation. This work was supported by grants from the Beijing Nova Program of Science and Technology (Z191100001119128); Beijing Municipal Science and Technology Project (Z191100006619081); National Natural Science Foundation of China (82073390); Beijing Municipal Administration of Hospitals' Youth Programme (QML20180108) and The Digestive Medical Coordinated Development Center of Beijing Municipal Administration of Hospitals (XXZ02, XXZ01). The study sponsors had no role in the design and preparation of this manuscript.

## Additional information

### Competing interests

Xiang Liu, Libo Zhao, Zhi Li: are employed by Echo Biotech Co, Ltd. The authors have no other competing interests to declare. The other authors declare that no competing interests exist.

### Funding

| Funder | Grant reference number | Author |
| --- | --- | --- |
| Beijing Nova Program | Z191100001119128 | Li Min |
| Beijing Municipal Science and Technology Commission, Adminitrative Commission of Zhongguancun Science Park | Z191100006619081 | Li Min |

| Funder | Grant reference number | Author |
|---|---|---|
| National Natural Science Foundation of China | 82073390 | Li Min |
| Beijing Municipal Administration of Hospitals | QML20180108 | Li Min |
| Beijing Municipal Administration of Hospitals | XXZ02 | Li Min |
| Beijing Municipal Administration of Hospitals | XXZ01 | Shengtao Zhu |

The funders had no role in study design, data collection and interpretation, or the decision to submit the work for publication.

### Author contributions

Li Min, Conceptualization, Formal analysis, Supervision, Funding acquisition, Writing – original draft, Writing – review and editing; Fanqin Bu, Software, Formal analysis, Visualization, Methodology; Jingxin Meng, Project administration; Xiang Liu, Software, Formal analysis, Methodology; Qingdong Guo, Data curation, Methodology; Libo Zhao, Formal analysis, Visualization; Zhi Li, Software; Xiangji Li, Visualization, Methodology; Shengtao Zhu, Supervision, Project administration; Shutian Zhang, Supervision, Investigation, Writing – review and editing

### Author ORCIDs

Li Min ⑩ https://orcid.org/0000-0001-9595-5536
Fanqin Bu ⑩ http://orcid.org/0009-0005-5006-9705

### Ethics

Clinical trial registration 2021-P2-230-01.
This study was approved by the ethics committee of Beijing Friendship Hospital, and written informed consent was obtained from each participant.

Joint Public Review: https://doi.org/10.7554/eLife.88675.4.sa1
Author response https://doi.org/10.7554/eLife.88675.4.sa2

# Additional files

## Supplementary files

- Supplementary file 1. CRC vs NC mRNA.
- Supplementary file 2. CRC vs NC miRNA.
- Supplementary file 3. CRC vs NC lncRNA.
- Supplementary file 4. AA vs NC mRNA.
- Supplementary file 5. AA vs NC miRNA.
- Supplementary file 6. AA vs NC lncRNA.
- Supplementary file 7. CRC vs AA mRNA.
- Supplementary file 8. CRC vs AA miRNA.
- Supplementary file 9. CRC vs AA lncRNA.
- Supplementary file 10. AA vs NC up-regulated Median_50 Log2FC_2 with Module sorted by FDR.
- Supplementary file 11. CRC vs NC up-regulated Median_50 Log2FC_2 with Module sorted by FDR.
- Supplementary file 12. Participants' characteristics for the training and validation cohorts.
- Supplementary file 13. Transcripts and sequence of their primers and probes.
- MDAR checklist

## Data availability

The RNA sequencing data have been deposited at the Sequence Read Archive (SRA) database of NCBI under the accession number PRJNA639943. This paper does not report original code. Any information required to reanalyze the data reported in this paper is available from the lead contact (minli@ccmu.edu.cn) upon request.

The following dataset was generated:

| Author(s) | Year | Dataset title | Dataset URL | Database and Identifier |
|---|---|---|---|---|
| Min L | 2024 | Circulating Small Extracellular Vesicle RNA Profiling for the Detection of T1a Stage Colorectal Cancer and Precancerous Advanced Adenoma | http://www.ncbi.nlm.nih.gov/bioproject/?term=PRJNA639943 | NCBI BioProject, PRJNA639943 |

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
