## [Editor Report · eLife assessment]

This study presents a **useful** description of RNA in extracellular vesicles (EV-RNAs) and highlights the potential to develop biomarkers for the early detection of colorectal cancer (CRC) and precancerous adenoma (AA). The data were analysed using overall **solid** methodology and would benefit from further validation of predicted lncRNAs and biomarker validation at each stage of CRC/AA to evaluate the potential application to early detection of CRC and AA.

---

## [Referee Report · Joint Public Review]

Detection of early-stage colorectal cancer is of great importance. Laboratory scientists and clinicians have reported different exosomal biomarkers to identify colorectal cancer patients. This is a proof-of-principle study of whether exosomal RNAs, and particularly predicted lncRNAs, are potential biomarkers of early-stage colorectal cancer and its precancerous lesions.

Strengths:

The study provides a valuable dataset of the whole-transcriptomic profile of circulating sEVs, including miRNA, mRNA, and lncRNA. This approach adds to the understanding of sEV-RNAs' role in CRC carcinogenesis and facilitates the discovery of potential biomarkers.

The developed 60-gene t-SNE model successfully differentiated T1a stage CRC/AA from normal controls with high specificity and sensitivity, indicating the potential of sEV-RNAs as diagnostic markers for early-stage colorectal lesions.

The study combines RNA-seq, RT-qPCR, and modelling algorithms to select and validate candidate sEV-RNAs, maximising the performance of the developed RNA signature. The comparison of different algorithms and consideration of other factors enhance the robustness of the findings.

Weaknesses:

Validation in larger cohorts would be required to establish as biomarkers and to demonstrate whether the predicted lncRNAs implicated in these biomarkers are indeed present and whether they are robustly predictive/prognostic.

The following points were noted during preprint review:

(1) Lack of analysis on T1-only patients in the validation cohort: While the study identifies key sEV-RNAs associated with T1a stage CRC and AA, the validation cohort is only half of the patients in T1(25 out of 49). It would be better to do an analysis using only the T1 patients in the validation cohort, so the conclusion is not affected by the T2-T3 patients.

(2) Lack of performance analysis across different demographic and tumor pathology factors listed in Supplementary Table 12. It's important to know if the sEV-RNAs identified in the study work better/worse in different age/sex/tumor size/Yamada subtypes etc.

(3) The authors tested their models in a medium size population of 124 individuals, which is not enough to obtain an accurate evaluation of the specificity and sensitivity of the biomarkers proposed here. External validation would be required.

(4) Depicting the full RNA landscape of circulating exosomes is still quite challenging. The authors annotated 58,333 RNA species in exosomes, most of which were lncRNAs, with annotation methods briefly described in Suppl Methods.

---

## [Author Response]

The following is the authors’ response to the previous reviews.

**Reviewer #1:**
Detection of early-stage colorectal cancer is of great importance. Laboratory scientists and clinicians have reported different exosomal biomarkers to identify colorectal cancer patients. This is a proof-of-principle study of whether exosomal RNAs, and particularly predicted lncRNAs, potential biomarkers of early-stage colorectal cancer and its precancerous lesions.Strengths:The study provides a valuable dataset of the whole-transcriptomic profile of circulating sEVs, including miRNA, mRNA, and lncRNA. This approach adds to the understanding of sEV-RNAs' role in CRC carcinogenesis and facilitates the discovery of potential biomarkers.The developed 60-gene t-SNE model successfully differentiated T1a stage CRC/AA from normal controls with high specificity and sensitivity, indicating the potential of sEV-RNAs as diagnostic markers for early-stage colorectal lesions.The study combines RNA-seq, RT-qPCR, and modelling algorithms to select and validate candidate sEV-RNAs, maximising the performance of the developed RNA signature. The comparison of different algorithms and consideration of other factors enhance the robustness of the findings.Weaknesses:Validation in larger cohorts would be required to establish as biomarkers, and to demonstrate whether the predicted lncRNAs implicated in these biomarkers are indeed present, and whether they are robustly predictive/prognostic.

Thank you for your careful evaluation and valuable suggestions, which have provided valuable guidance for the improvement of our paper. In response to your feedback, we have implemented the following improvements.

(1) More detail about how lncRNA and miRNA candidates were defined, and how this compares to previously published miRNA and lncRNA predictions. The Suppl Methods section for lncRNAs does not describe in detail how the "CPC/CNCI/Pfam" "methods" were combined to define lncRNAs here.

Author response and action taken: Thanks for your comments. In the Supplementary Methods section titled " Selection of Predictive Biomarkers", we have provided a more detailed illustration regarding the screening process for candidate RNA biomarkers. The revised section is as follows: To ensure the predictive performance of the sEV-RNA signature, candidate sEV-RNAs were ultimately selected based on their fold change in colorectal cancer/ precancerous advanced adenoma, absolute abundance, and module attribution. In detail, we initially selected the top 10 RNAs from each category (mRNA, miRNA, and lncRNA) with a fold change greater than 4. In cases where fewer than 10 RNAs were meeting this criterion, all RNAs with a fold change greater than 4 were included. Subsequently, we filtered out RNAs with low abundance, and we selected the top-ranked RNAs from each module based on the fold change ranking for inclusion in the final model.

Compared to most previous studies on EV biomarkers, the overall discriminative performance of the biomarker model we constructed is considerable, holding clinical value for practical application. In contrast, the supplementary merit of this study lies in uncovering the heterogeneity at the whole transcriptome level among samples of different categories, providing a more comprehensive insight into the dynamic changes of biological states. For instance, we inferred the cell subtypes of EV origins through ssGSEA and correlated them with the tumor microenvironment status. The regulatory relationships among different RNA categories were delineated, and their impacts on biological signaling pathways were analyzed, a feat challenging to accomplish solely through sequencing of a single RNA category.

In the Supplementary Methods section titled " Identification of mRNAs and lncRNAs", we have provided a more detailed explanation regarding how the "CPC/CNCI/Pfam" methods were combined to define lncRNAs. The revised section is as follows: Three computational approaches including CPC (Coding Potential Calculator)/CNCI (Coding-Non-Coding Index)/Pfam were combined to sort non-protein coding RNA candidates from putative protein-coding RNAs in the unknown transcripts. CPC is a sequence alignment-based tool used to assess protein-coding capacity. By aligning transcripts with known protein databases, CPC evaluates the biological sequence characteristics of each coding frame of the transcript to determine its coding potential and identify non-coding RNAs.1 CNCI analysis is a method used to distinguish between coding and non-coding transcripts based on adjacent nucleotide triplets. This tool does not rely on known annotation files and can effectively predict incomplete transcripts and antisense transcript pairs.2 Pfam divides protein domains into different protein families and establishes statistical models for the amino acid sequences of each family through protein sequence alignment.3 Transcripts that can be aligned are considered to have a certain protein domain, indicating coding potential, while transcripts without alignment results are potential lncRNAs. Putative protein-coding RNAs were filtered out using a minimum length and exon number threshold. Transcripts above 200 nt with more than two exons were selected as lncRNA candidates and further screened by CPC/CNCI/Pfam. We distinguished lncRNAs from protein-coding genes by intersecting the results of the three determination methods mentioned above.

(2) The role and function of many lncRNAs are unknown, and some lncRNA species may simply be the product of pervasive transcription. Although this is an exploratory and descriptive study of potential biomarkers, it would benefit from some discussion of potential mechanisms because the proposed prediction models include lncRNAs. Do the authors have a hypothesis as to why lncRNAs were informative and predictive in this study? Are these lncRNAs well-studied and/or known to be functional? Or are they markers for pervasive transcription, for example?

Author response and action taken: Thanks for your comments. Whole transcriptome sequencing results facilitate the discussion of regulatory mechanisms between different biomarkers, supplying evidence for future investigations. Among the three lncRNAs involved in this study, lnc-MKRN2-42:1 is involved in the occurrence and development of Parkinson's disease4. The other two lncRNAs, however, lack relevant reports. Therefore, we cannot confirm that these lncRNAs have specific biological functions. In the Supplementary Methods section titled " Identification of mRNAs and lncRNAs", we acknowledge the limited understanding of sEV-lncRNAs in current research. In contrast, many miRNAs in the model have been proven to participate in the occurrence and development of colorectal cancer, such as miR-36155, miR-425-5p6, and miR-106b-3p7. These data provide biological support for the performance of the model, which is particularly valuable for model prediction.

(3) In the Results section "Cell-specific features of the sEV-RNA profile indicated the different proportion of cells of sEV origin among different groups", the sEV-RNA profiles were correlated with existing transcriptome profiles from specific cell types (ssGSEA) and used to estimate "tumour microenvironment-associated scores". This transcriptomic correlation is a valuable observation, but there is no further evidence provided that the sEV-RNAs profiles truly reflect differential cell types of sEV origin between the sample subgroups.Could the authors clarify the strength of evidence for the cells-of-origin estimates, which are based only on sEV-RNA transcriptome profiles? Would sEV-RNA-derived cells-of-origin be expected to correlate with histopath-derived scores (tumour microenvironment; immune infiltrate) for example? Or is this section intended as an exploratory description of sEV-RNAs, perhaps a check on the plausibility of the sEV-RNA profiles, rather than an accurate estimation of cells-of-origin in each subgroup?

Author response: Thanks for your comments. This section explores the proportional distribution of EVs from different cellular subgroups solely based on transcriptome profiles and algorithms, rather than providing precise estimates of cellular origins within each subgroup.

(4) Software and R package version numbers should be provided.

Author response and action taken: Thanks for your comments. We have added version information for relevant R packages at the first mention in the original text (e.g., WGCNA (version 1.61), Rtsne (version 0.15), GSVA (version 1.42.0), ESTIMATE (version 1.0.13), DOSE (version 3.8.0)).

References

(1) Kong L, et al. CPC: assess the protein-coding potential of transcripts using sequence features and support vector machine. Nucleic Acids Res. 35, W345-349 (2007).

(2) Sun L, et al. Utilizing sequence intrinsic composition to classify protein-coding and long non-coding transcripts. Nucleic Acids Res. 41, e166 (2013).

(3) Finn RD, et al. Pfam: the protein families database. Nucleic Acids Res. 42, D222-230 (2014).

(4) Wang Q, et al. Integrated analysis of exosomal lncRNA and mRNA expression profiles reveals the involvement of lnc-MKRN2-42:1 in the pathogenesis of Parkinson's disease. CNS Neurosci Ther. 26, 527-537 (2020).

(5) Zheng G, et al. Identification and validation of reference genes for qPCR detection of serum microRNAs in colorectal adenocarcinoma patients. PLoS One. 8, e83025 (2013).

(6) Liu D, Zhang H, Cui M, Chen C, Feng Y. Hsa-miR-425-5p promotes tumor growth and metastasis by activating the CTNND1-mediated β-catenin pathway and EMT in colorectal cancer. Cell Cycle. 19, 1917-1927 (2020).

(7) Liu H, et al. Colorectal cancer-derived exosomal miR-106b-3p promotes metastasis by down-regulating DLC-1 expression. Clin Sci (Lond). 134, 419-434 (2020).